# Physiological and Transcriptome Analysis of the Effects of Exogenous Strigolactones on Drought Responses of Pepper Seedlings

**DOI:** 10.3390/antiox12122019

**Published:** 2023-11-21

**Authors:** Huangying Shu, Muhammad Ahsan Altaf, Naveed Mushtaq, Huizhen Fu, Xu Lu, Guopeng Zhu, Shanhan Cheng, Zhiwei Wang

**Affiliations:** 1Key Laboratory for Quality Regulation of Tropical Horticultural Crops of Hainan Province, School of Breeding and Multiplication (Sanya Institute of Breeding and Multiplication), Center of Nanfan and High-Efficiency Tropical Agriculture, Hainan University, Sanya 572025, China; shuhuangying@hainanu.edu.cn (H.S.); ahsanaltaf@hainanu.edu.cn (M.A.A.); 2015204041@njau.edu.cn (N.M.); fhzhenz@hainau.edu.cn (H.F.); luxu@hainanu.edu.cn (X.L.); zhuguopeng@hainanu.edu.cn (G.Z.); 990865@hainanu.edu.cn (S.C.); 2Key Laboratory for Quality Regulation of Tropical Horticultural Crops of Hainan Province, School of Tropical Agriculture and Forestry, Hainan University, Haikou 570228, China; 3Hainan Yazhou Bay Seed Laboratory, Sanya 572025, China

**Keywords:** pepper, drought stress, strigolactone, transcriptome, molecular

## Abstract

Drought stress significantly restricts the growth, yield, and quality of peppers. Strigolactone (SL), a relatively new plant hormone, has shown promise in alleviating drought-related symptoms in pepper plants. However, there is limited knowledge on how SL affects the gene expression in peppers when exposed to drought stress (DS) after the foliar application of SL. To explore this, we conducted a thorough physiological and transcriptome analysis investigation to uncover the mechanisms through which SL mitigates the effects of DS on pepper seedlings. DS inhibited the growth of pepper seedlings, altered antioxidant enzyme activity, reduced relative water content (RWC), and caused oxidative damage. On the contrary, the application of SL significantly enhanced RWC, promoted root morphology, and increased leaf pigment content. SL also protected pepper seedlings from drought-induced oxidative damage by reducing MDA and H_2_O_2_ levels and maintaining POD, CAT, and SOD activity. Moreover, transcriptomic analysis revealed that differentially expressed genes were enriched in ribosomes, ABC transporters, phenylpropanoid biosynthesis, and Auxin/MAPK signaling pathways in DS and DS + SL treatment. Furthermore, the results of qRT-PCR showed the up-regulation of *AGR7*, *ABI5*, *BRI1*, and *PDR4* and down-regulation of *SAPK6*, *NTF4*, *PYL6*, and *GPX4* in SL treatment compared with drought-only treatment. In particular, the key gene for SL signal transduction, *SMXL6*, was down-regulated under drought. These results elucidate the molecular aspects underlying SL-mediated plant DS tolerance, and provide pivotal strategies for effectively achieving pepper drought resilience.

## 1. Introduction

Plants are subjected to various stresses from their surrounding environment throughout their entire life. Drought stress (DS) is becoming more prominent as an abiotic stress factor that affects the yield of horticultural crops. In recent decades, drought intensity, frequency, and severity have posed a severe threat to worldwide agricultural production [1]. However, drought tolerance in plants is a multifaceted characteristic that is significantly influenced by environmental factors. DS significantly altered the morphological, physiological, and metabolic functions in plants [2]. Plant tolerance mechanisms for DS may give genetic resources to generate drought-resistant crops.

Pepper (*Capsicum* spp.) is one of the world’s major vegetable crops, and is widely cultivated due to its exceptional adaptability [3]. Additionally, peppers are rich source of vitamins, pigments, carbohydrates, and antioxidants [4]. The continuing deterioration of the climate, the frequent occurrence of high temperatures, and the shortage of water during the pepper growth and development period have a major impact on pepper quality and output. In light of this, it is crucial to understand the physiological and molecular responses of chili peppers to DS and enhance their capacity for resilience. Strigolactone (SL) is a novel class of plant hormone, characterized by a diverse array of chemical structures. It is a sesquiterpenoid lactone that is generated from a tetracyclic carotenoid. Its primary function is to inhibit plant branching [5,6]. The use of phytohormones in the agricultural sector suggests its potential for enhancing drought tolerance, increasing crop production, and regulating plant growth. SL is a promising tool for improving both crop productivity and resilience in plants [7]. Exogenous SL can alleviate the adverse effects of DS by regulating stomatal closure, chlorophyll synthesis, and photosynthesis, as well as activating the antioxidant defense protection mechanism in grapevines [8]. Importantly, exogenous SL efficiently improved the sensitivity of stomata to abscisic acid (ABA) in a DS environment [9]. In *Arabidopsis*, SL acts as a positive regulator in the regulation of the plant’s response to DS. Comparative transcriptome investigations have demonstrated that plants effectively integrate several hormone-signaling pathways, such as SL, ABA, and cytokinin, to cope with environmental stresses [10]. *SsMAX2* has been reported to reduce chlorophyll degradation and water loss, thereby decreasing H_2_O_2_ levels in *Arabidopsis* [11]. Although SL’s regulation of plant stress responses is well documented, its specific impact on enhancing DS resistance in pepper has received comparatively limited attention. Consequently, there is an urgent need to comprehensively explore SLs and their uses, and investigate their potential applications in pepper seedlings under DS conditions.

SL has the potential to serve as an effective approach for mitigating the adverse effects of drought stress on plants. The molecular pathways involved in regulating drought tolerance via exogenous SL, especially when pepper seedlings are exposed to drought conditions, have not yet been determined. The investigation of the role of SL in the mitigation of DS in pepper is an intriguing avenue for research. In order to address these concerns, we conducted an analysis of the phenotypic, physiological, biochemical, and molecular pathways of pepper seedlings under DS. Furthermore, we explored the potential mechanisms of SL in alleviating DS in pepper seedlings and identify genes associated with DS responses.

## 2. Materials and Methods

### 2.1. Plant Materials and Treatments

*Capsicum chinense* (HNUCC16) was cultivated in growth chambers under control conditions (16/8 h day/night photoperiod, 26/22 °C day/night temperature, and relative humidity of 75%). At the 6–8 leaf stage, identical seedlings were transferred into plastic containers containing Hoagland’s nutrient solution (HNS). HNS was replaced every three days. After a 7-day adaptation period, the plants were separated into different treatment groups, as follows: (1) control; (2) drought stress (DS), in which plants were placed in 10% (*w*/*v*) polyethylene glycol (PEG-6000) to simulate drought conditions; and (3) 3 µM GR24 pretreatment, in which plants were placed in 10% (*w*/*v*) PEG-6000 (SL + DS). Each treatment group included 15 plants. GR24 (Coolaber, Beijing, China), an SL analog, was dissolved with acetone and 0.1% Tween 20, and the storage concentration was 10 mM. Control and DS were then treated with the equivalent amount of distilled water with the same amount of acetone and Tween 20. SL was applied to the leaves on peppers for three consecutive days (twice a day). After SL application, seedlings were exposed to drought stress treatment for three days. After stress treatment, leaf samples were collected for morphology, physiological, and biochemical assays and RNA sequencing analysis. All samples were immediately frozen in liquid nitrogen and stored at −80 °C. Each treatment was repeated three times. Pepper seeds were supplied by the School of Tropical Agriculture and Forestry, Hainan University (Hainan, China). Three biological replicates were used in each treatment.

### 2.2. Measurement of Relative Water Content and Root Morphology

The relative water content (RWC) was measured using a modified protocol described previously [8]. The pepper leaves were rinsed with distilled water, and leaves were wiped with absorbent paper and the fresh weight (FW) recorded. The samples were soaked in distilled water for 24 h at 4 °C in darkness, and the turgid weight (TW) was recorded. The dry weight (DW) was then determined after drying at 85 °C for 2 min and 24 h at 60 °C. RWC formula:RWC (%) = (FW − DW)/(TW − DW) × 100%

Roots’ phenotype was recorded after seven days of DS. To examine the root morphology, four plants of similar size were chosen from each duplicate, and their roots were taken. Subsequently, the roots were completely cleaned with running tap water. An Image Scanning Screen (Epson Expression 110000XL, Regent Instruments, Québec, QC, Canada) was used for root scanning, and WinRHIZO 2003a software was used for root data analysis.

### 2.3. Scanning Electron Microscopy

The leaf tissue blocks were left in an electron microscopy fixative for 2 h at room temperature, then transferred to 4 °C for preservation and transport. Then, leaf tissue blocks were treated with 0.1 M PB (pH 7.4). Then, tissue blocks were transferred to 1% OsO_4_ in 0.1 M PB (pH 7.4). After that, leaf tissue blocks were infused with alcohol and isoamyl acetate. Dry samples were dried with a dryer, and specimens were attached to metallic stubs using carbon stickers and sputter coated with gold. Finally, we observed and captured images with a scanning electron microscope (SU8100, Hitachi, Tokyo, Japan).

### 2.4. Chlorophyll Content

For the determination of pigment molecules, 0.1 g fresh leaves ground, and placed in a 10 mL of acetone extracting solution. The sample was centrifuged at 10,000× *g* for 10 min. The chlorophyll was spectrophotometrically determined in supernatant at 663 and 645 nm, respectively. Determination of chlorophyll content was realized using the formulae reported by [12].

### 2.5. Measurement of Antioxidant Enzyme Activity and Related Metabolites

The activities of superoxide dismutase (SOD), peroxidase (POD), and catalase (CAT) were analyzed based on the modified protocols reported by [12]. Fresh samples (0.1 g) were ground in 900 µL phosphate buffer (pH 7.8), and the homogenate was centrifuged at 10,000× *g* for 10 min. The supernatant was used to determine the activity of antioxidant enzymes. The MDA level, H_2_O_2_ contents, the Pro concentration, and the soluble sugar content of the leaves were determined using an assay kit (Jiancheng Bioengineering Institute, Nanjing, China), following the detailed instructions [12]. The colorimetric measurements of MDA, H_2_O_2_, Pro, and soluble sugar were measured at 530, 405, 520, and 620 nm, respectively.

### 2.6. Transcriptome Sequencing, Differential Gene Expression, and Enrichment Analysis

We selected explants with normal development for sampling in different treatments, including control, DS, and SL+DS. Total RNA was isolated with Trizol Reagents (Thermo Fisher Scientific, Shanghai, China). Nine nondirectional libraries were produced using the NEBNext^®^ UltraTM RNA Library Prep Kit for Illumina^®^ (NEB, Boston, Massachusetts, USA) and sequenced on the Illumina Novaseq platform. Clean reads were obtained by filtering low-quality reads and aligning them with the *C. chinense* reference genome using HISAT2 (v.2.0.5) [13]. The methods of Benjamini and Hochberg were used to adjust the obtained *p*-value to control the error detection rate. Genes with an adjusted *p*-value < 0.05 discovered through DESeq2 were assigned as differentially expressed (DEG) [14]. Each subset of the clustering line graph, where the value of differential genes is the union of all comparative combinations of differential genes, and the expression level of differential genes in the FPKM expression matrix of each sample were taken as log2 (fpkm + 1) and centralized for correction. The Gene Ontology (GO) of differentially expressed genes and the statistical enrichment of DEGs in the KEGG pathway were analyzed using the clusterProfiler R package [15,16].

### 2.7. Gene Expression by qRT-PCR

Total RNA was reversed transcriptionally using Hiscript Q RT SuperMix for qPCR (Vazyme Biotech, Nanjing, China), and qPCRs were analyzed using ChamQ SYBR qPCR Master Mix (Vazyme Biotech, Nanjing, China). The amplification program was based on the QuantStudio^TM^ 5 Real-Time PCR System (Thermo Fisher Scientific, Waltham, Massachusetts, USA) [17]. The specific primers of the selected genes are listed in Appendix A.

### 2.8. Statistical Analysis

Statistical analysis was performed using Microsoft Excel (v16.72) and SPSS (v26). The significance of the difference between the mean values was tested using Duncan’s multiple range test (*p* < 0.05). Figures were generated using GraphPad Prism software (v9.5.0), representing mean values with a standard deviation of three replicates per treatment.

## 3. Results

### 3.1. SL Alleviated the Negative Effects of Drought Stress on the Morphology of Pepper

To confirm the possible effects of SL on pepper growth in a DS environment, we examined the pepper morphology in the control, DS, and DS + SL groups. DS significantly inhibited the root growth of pepper seedlings. However, pretreatment with SL can effectively alleviate the inhibitory effect of DS. Under DS conditions, the root length, surface area, volume, tips, crossing, and forks of the plant decreased by 49.56%, 59.45%, 65.27%, 32.31%, 55.7%, and 56.62%, respectively, compared with the control treatments. Furthermore, after SL pretreatment, the reduction was only 20.13%, 35.48%, 35.71%, 15.4%, 29.11%, and 22.63%, respectively, compared to the control group. Importantly, foliar application of SL significantly promoted the root architecture system of pepper seedlings in a DS environment (Figure 1). These findings suggested that SL could maintain normal root growth and balanced water uptake.

### 3.2. The Microscopic Structure of Leaf Tissue

We evaluated the stomatal characteristics of plants with various treatments to detect the effect of SL on their stomatal opening and closing ability under DS (Figure 2). SEM showed that compared to the control group, the plant stomata were completely closed in drought-only treatment (Figure 2B). In SL application along with DS conditions, stomatal closure was alleviated, demonstrating that exogenous SL could alter the stomata opening in response to DS (Figure 2C). Additionally, compared to the control group, the stomatal area decreased by 98%; however, the stomatal area increased by 57.45% with SL foliar application under drought stress (Figure 2D). SL foliar application may promote stomatal opening by regulating cell permeability to maintain water levels in leaves.

### 3.3. Relative Water Content and Chlorophyll Content of the Leaves

The drought strongly affected the RWC and chlorophyll content of the pepper seedlings. Only DS treatment has a lower RWC (76.57%) than the control group (90.61%). Remarkably, peppers treated with DS + SL have a significantly higher RWC in leaves (86.35%) than those grown under drought-only conditions (Figure 2E). Drought treatment significantly reduced the chlorophyll content in the leaves of pepper seedlings. Compared to the control, the concentrations of total chlorophyll, chlorophyll a, and chlorophyll b in pepper seedlings were decreased by 28.8%, 26.82%, and 21.91%, respectively. However, DS + SL plants exhibit considerably higher chlorophyll content, with increases of 23.69%, 22.33%, and 12.78%, respectively, compared to DS treatment plants (Figure 2F–H).

### 3.4. Measurement of Antioxidant Enzyme Activity and Related Metabolites

Drought + SL treatments significantly affected the antioxidant enzyme (POD, CAT, and SOD) activity of pepper seedlings. DS had significantly increased POD, CAT, and SOD activity compared to the control group. However, pretreated SL inhibited enzyme activity in the leaves of pepper seedlings. For example, the POD, CAT, and SOD activity of DS was only 180.86%, 93.59%, and 107.02% higher than in the control group. In contrast, the DS + SL group had lower enzyme activity than the DS-only treatments (Figure 3A–C). Compared to the control group, the DS-only group presented an extreme increase in proline, MDA, and H_2_O_2_. On the contrary, DS + SL plants presented significantly decreased proline, MDA, and H_2_O_2_ content, with 88.39%, 57.89%, and 61.83%, respectively, compared with the DS group in pepper seedlings (Figure 3D–F). Under DS, there is an increase in soluble sugar content. In addition, no significant differences were seen in DS and DS + SL treatments (Figure 3G).

### 3.5. Assessment of RNA-Seq Data and Differentially Expressed Gene Analysis

A total of 40.32 million raw data were generated from transcriptome sequencing in three groups, which were filtered for low-quality reads, resulting in 38.69 million clean reads. It was found that 88.58–93.36% of reads could be successfully assigned to the *C. chinense* reference genome, and the unique mapping rate ranged from 85.35–89.02% (Appendix A). Hierarchy analysis showed that different samples at the same treatment clustered together, and the DS + SL group demonstrated the most significant differences (Figure 4A). The transcriptome was divided into four clusters of DEGs, and the genes in each cluster had similar expression patterns, including 95, 1481, 159, and 25, respectively (Figure 4B). A total of 2097 DEGs were identified among the DS vs. control, DS + SL vs. control, and DS vs. DS + SL groups. The Venn diagram showed that only two genes were expressed in all groups, and the specific DEGs were higher than the common DEGs among all three groups (Figure 4C). Furthermore, there were 410 and 849 DEGs in group DS and DS + SL, compared with the control group, respectively. Moreover, there were 838 DEGs between the DS and DS + SL groups (Figure 4D).

### 3.6. Enrichment Analysis of DEGs

Comparing DEGs between the control and DS group, galactose metabolism, plant hormone signal transduction, photosynthesis, and photosynthesis-antenna proteins were enriched in the latter. When control and DS + SL were compared, it was found that the biosynthesis of amino acids, brassinosteroid biosynthesis, carotenoid biosynthesis, and MAPK signaling were enriched in the latter. Intriguingly, the DS and DS + SL groups exhibited a significant enrichment in the ribosome pathway compared to the control group. Phenylpropanoid biosynthesis, terpenoid backbone biosynthesis, flavonoid biosynthesis, carotenoid biosynthesis, and MAPK signaling pathways were enriched in the DEGs identified between treatments with DS and DS + SL (Figure 5 and Appendix A and Appendix A). In addition, the genes of interest, as described by the molecular and biological functions of GO, were classified as molecular function (MF), cellular component (CC), and biological process (BP). GO functional annotation was performed to fully understand the roles of DEGs among the control, DS, and DS + SL groups (Appendix A).

### 3.7. Candidate Genes Involved in Drought and SL Treatment

Based on enrichment analysis, we identified 71 genes whose expression showed significant differences among the control, DS, and DS + SL groups, signifying that they may be responsive to DS or involved in the SL-induced alleviation of DS. Genes associated with the phenylpropanoid biosynthesis pathway were considerably enriched between DS alone and the DS + SL group, and 18 differentially expressed genes were identified. Interestingly, the expression of the genes related to phenylpropanoid biosynthesis *BC332_05436* (cytochrome 84A1) and *BC332_01431*(*peroxidase 15*) were notably higher in DS + SL group than in DS group plants. On the contrary, *BC332_25211*(*4-coumarate*--*CoA ligase 2*), *BC332_33581*(*caffeoyl-CoA O-methyltransferase*), and *BC332_34596* (*caffeoyl-CoA O-methyltransferase*) show substantially lower expression in DS + SL than in drought-only treatment. Furthermore, compared with the control group, in plants under DS alone and DS + SL treatment, genes related to the plant hormone signal transduction and MAPK pathways were significantly enriched. *BC332_22503* (*indole-3-acetic acid-induced protein ARG7*) and *BC332_09592* (*BRI1 kinase inhibitor 1*) were distinctly expressed in the DS + SL group, and had low expression in DS. Additionally, we identified 15 SL biosynthesis-related genes from DEGs. The expression profiles of genes involved in SL biosynthesis showed significant differences among the control, DS, and DS + SL groups. Compared with the DS group, *BC332_23620* (*SMAX1-LIKE 6*), *BC332_15224* (*SMAX1-LIKE 7*), and *BC332_18735* (*SMAX1-LIKE 4*) were expressed higher in DS + SL. Furthermore, 23 genes were identified in the ABC transporter pathway. Compared to the control treatment, the genes *BC332_30010*, *BC332_33666*, and *BC332_16040* were up-regulated in DS + SL treatment but down-regulated in DS-treated plants (Figure 6). Furthermore, we acquired mRNA levels of proline and antioxidant enzyme biosynthetic genes, which further elucidated the candidate genes at the molecular level (Appendix A).

### 3.8. qRT-PCR Validation of Gene Expression

Fifteen candidate genes closely associated with DS were selected for analysis of expression via RNA-Seq (Appendix A). The control group was used as a control, while the DS and DS + SL groups were investigated to validate the reliability of the RNA-seq data obtained in this study (Figure 7).

## 4. Discussion

Drought is well recognized as a primary abiotic stress factor that impedes plant growth and development [18]. The crucial role of several plant hormones in regulating the response to plant stress is well recognized [19]. SL has a crucial function in the regulation of plant development and the alleviation of environmental stressors [20]. Furthermore, the foliar application of SL enhances plant growth under DS conditions by improving photosynthetic characteristics and antioxidant enzyme activity [21]. However, little information about its effects on pepper abiotic stress responses is available.

DS considerably inhibited plant growth [22]. In this study, the roots and leaves of pepper seedlings were adversely affected under DS conditions. Pretreatment with SL can alleviate the drought symptoms of peppers. In the drought-stress treatment, the young wilted leaves showed signs of drooping and shrinking and most mature leaves had begun to curl, whereas the leaves of the DS + SL treatment only showed mild symptoms (Figure 1 and Figure 2E). Hence, the results indicated that the application of SL reduced the apparent drought damage of pepper seedlings under DS. Chlorophyll content in plants is a critical indicator of photosynthetic activity, but this characteristic is adversely affected under stressful conditions [23]. The chlorophyll content significantly reduced by DS [24]. Additionally, SL plays a vital role in regulating plant photosynthetic efficiency [25]. Our research results show that the exogenous application of SL significantly alleviates the decrease in chlorophyll content under DS (Figure 2F–H). Consequently, this suggests that SL is an effective substance with which pepper can cope with DS; similar results have been reported in grapes [26]. Thus, applying SL is very possibly a method that can be used to alleviate the adverse effects of DS on pepper seedlings.

Proline is considered an important metabolite synthesized within plant cells under environmental stress conditions. It is assumed to have a significant impact on the protective function of plants grown under stress, which is attributed to its ability as an osmotic protector, membrane stabilizer, and ROS scavenger [27,28]. Previous literature suggested that proline accumulation may help improve water status and reduce oxidative damage caused by abiotic stress [18]. Similarly, we observed that DS significantly increased proline accumulation in pepper plants. In contrast, the application of SL significantly reduced the proline concentration in pepper seedlings under DS (Figure 3D). This demonstrates that SL has a potential efficiency effect on osmotic regulation during DS, which may help plants maintain growth and function. The proline biosynthesis gene (*AtP5CS1*) is pivotal in increasing proline biosynthesis under abiotic stress conditions [29]. We found that the mRNA level of *P5CS* (*BC332_15138*) was increased in peppers exposed to drought with or without SL (Appendix A). Captivatingly, the application of SL resulted in a significant decrease in proline content, which demonstrated that the up-regulation of *P5CS* transcription levels might contribute to an increase in proline concentration. However, the mechanism by which SL affects proline is not yet clear, and requires further exploration.

Maintaining redox homeostasis is of the utmost importance in mitigating the excessive generation of ROS and minimizing cellular membrane impairment in plants subjected to environmental stressors [30]. In the present study, there was a significant increase in the content of H_2_O_2_ and MDA under DS, while SL reversed the oxidative damage in DS-induced pepper seedlings by inhibiting the accumulation of H_2_O_2_ and MDA content (Figure 3E,F). The SL-mediated drought response may involve a complex interaction in H_2_O_2_ content and stomatal closure [8,31]. It may be necessary for plants to maintain the expression of some genes to maintain stomatal opening at specific levels, and subsequently balance CO_2_ inflow and water loss under water scarcity conditions [32]. We perceived that the leaf stomata were firmly closed during DS. However, SL application had a remarkable effect in relieving this stomatal closure (Figure 2C), suggesting that SL plays a crucial role in regulating the plant’s water balance, leading to the reopening of the stomata under DS. This is a complex molecular mechanism that still needs further research. To cope with oxidative damage, plants have developed a proficient antioxidant defense mechanism [33]. It was stated that SL can enhance the activity of antioxidant enzymes for ROS detoxification, thus endowing crop tolerance [34]. Compared with the DS group, lower levels of POD, CAT, and SOD activity were detected in the DS + SL group, signifying that SL can improve ROS clearance efficiency and potentially protect plants from the adverse effects of DS, consequently alleviating the oxidative stress caused by DS (Figure 3A–C). Furthermore, we assessed the expression levels of *peroxidase4* (*POD4*) and *peroxidase* (*POD15*), which are genes involved in the phenylpropane biosynthetic pathway. In drought-exposed seedlings, the expression of *POD15* decreased, whereas SL application significantly increased the expression levels of these genes, consistent with the transcriptomic results. Conversely, *POD4* showed almost no expression after SL treatment (Figure 7). Whether the transcription levels of *POD* genes contribute to improved drought tolerance or hinder drought resistance warrants further investigation.

SL effectively regulates plant growth in responses to various environmental conditions [35]. The physiological and molecular analysis of *Arabidopsis* suggests that plant hormone signaling pathways are critical to regulating drought or water-deficit responses [36]. *OSRK1* (*SAPK6*) is an upstream regulatory factor for stress signaling in rice roots, playing a significant role in ABA and hypertonic stress signaling [37]. Our research found that plants pretreated with SL exhibited significant gene changes associated with plant hormone signal transduction and the MAPK signal transduction pathway during DS (Figure 5 and Figure 6). Specifically, we observed changes in the expression of auxin-responsive proteins from the Aux/IAA family, that are known to play a key role in plant stress responses [38]. An increase in ROS concentration exerts a negative regulatory influence on IAA turnover [39]. Interestingly, in this study, the expression level of *ARG7* (*BC332_22503*) was found to be decreased by 97.9% in the DS group, while it significantly increased in the DS + SL group. Furthermore, *BC332_20130* (*SAPK6*) encoding kinase was found to be up-regulated in response to DS without SL treatment, but significantly down-regulated in the group with SL pretreatment. We hypothesize that these genes may mitigate DS through the action of SL.

ABC transporters are now recognized to be involved in many physiological processes that enable plants to adapt environmental changes [40]. Multipotent drug-resistance (PDR) subfamily ABC proteins are found in many plants, and cope with biotic and abiotic stresses. A previous study demonstrated that the overexpression of the ABC transporter protein *AtABCG36/AtPDR8* made plants more resistant to drought and salt stress than wild-type plants. In contrast, knockout lines are more sensitive to DS than wild-type plants [41]. We identified that *BC332_16040* (PDR4) was significantly up-regulated after SL pretreatment under DS, implying that *PDR4* may play a crucial role in the response to DS and SL. Previous studies have demonstrated that the interaction between *MAX2* and *DWARF14* triggers the signal transduction of SL, while *SMXL6* (*SUPPRESSOR OF MAX1-Like 6*) functions as an inhibitor of SL signal transduction [42,43]. The *MAX2* mutant exhibits high sensitivity to DS [44]. A recent study has discovered that *SsMAX2* overexpression in *Arabidopsis* significantly enhances resistance to drought, osmotic, and salt stresses [11]. In addition, *SMXL6*, *SMXL7*, and *SMXL8* play a negative regulatory role during DS in *Arabidopsis* [45]. We found that the expression level of *BC332_17885* (*CcSMAX1*) was up-regulated under DS but down-regulated with SL treatment (Figure 7), indicating its negative regulatory role in response to DS. On the contrary, the expression of *CcSMXL6* showed the opposite trend. The investigation of the mechanism of action is elusive, necessitating further investigation.

## 5. Conclusions

In summary, our comprehensive investigation, which includes phenotypic observations, physiological measurements, and gene expression analysis, demonstrates the multifaceted positive impact of SL on pepper plants in a DS environment. SL effectively mitigates the adverse effects of DS and promotes the growth of pepper seedlings under such challenging conditions. The application of SL is a powerful tool to counteract drought-induced oxidative damage and enhance tolerance to DS, ultimately facilitating the overall healthy development of pepper plants. Furthermore, our findings highlight that DEGs critical to DS responses are predominantly related to the MAPK and plant hormone signal transduction pathways. These insights contribute to a deeper understanding of the molecular and physiological mechanisms underlying the ability of pepper to withstand DS and provide a valuable conceptual framework for elucidating how SL intricately interacts with diverse signaling pathways in pepper’s adaptive response to drought stress. This research lays a solid foundation for future investigations into enhancing crop resilience and sustainability in the face of environmental challenges (Figure 8).

## Figures and Tables

**Figure 1 antioxidants-12-02019-f001:**
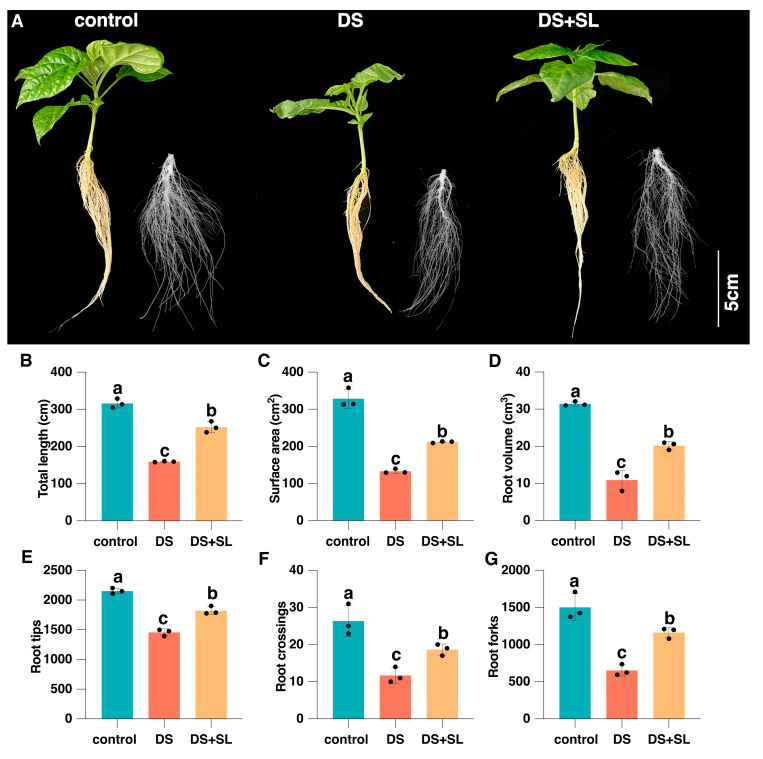
Phenotypic changes in pepper seedlings. (**A**) The phenotypic of pepper plants and roots under different treatments. (**B**) Total root length, (**C**) root surface area, (**D**) root volume, (**E**) root tips, (**F**) root crossing, and (**G**) root forks. Control: under normal control conditions, DS: drought conditions, and DS + SL: drought conditions with SL application. The data are mean values and standard errors; the black dot indicates three biological replicates. Different letters indicate significant differences between different treatments (*p* < 0.05).

**Figure 2 antioxidants-12-02019-f002:**
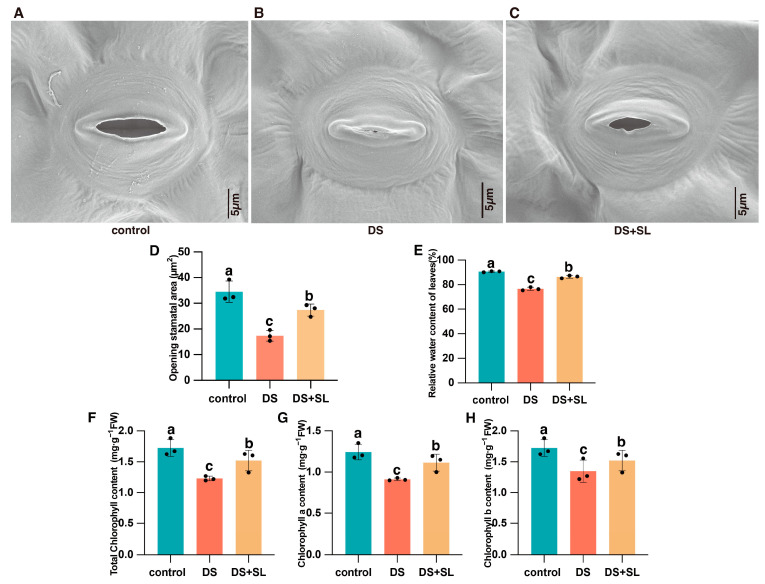
The combined effect of stomatal aperture performance and physiological analyses of pepper plants under control conditions, DS (drought conditions), and DS + SL (drought conditions with SL application). (**A**–**C**) Stomatal aperture performance; (**D**) analysis of stomatal area in different groups; (**E**) relative water content determined; (**F**) the total chlorophyll of the leaves; (**G**) chlorophyll a of the leaves; and (**H**) chlorophyll b of the leaves. The data are mean values and standard errors; the black dot indicates three biological replicates. Different letters indicate significant differences between different treatments (*p* < 0.05).

**Figure 3 antioxidants-12-02019-f003:**
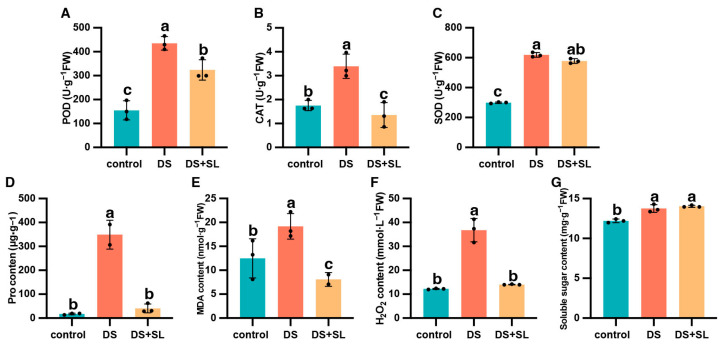
Effects of enzymatic and non-enzymatic antioxidant content in control, DS, and DS + SL pepper plant leaves. The antioxidant enzyme activity of (**A**) POD, (**B**) CAT, and (**C**) SOD; (**D**) Proline content, (**E**) MDA content, (**F**) H_2_O_2_ content, and (**G**) Soluble sugar content. The data are mean values and standard errors; the black dot indicates 3 biological replicates. Different letters indicate significant differences between different treatments (*p* < 0.05).

**Figure 4 antioxidants-12-02019-f004:**
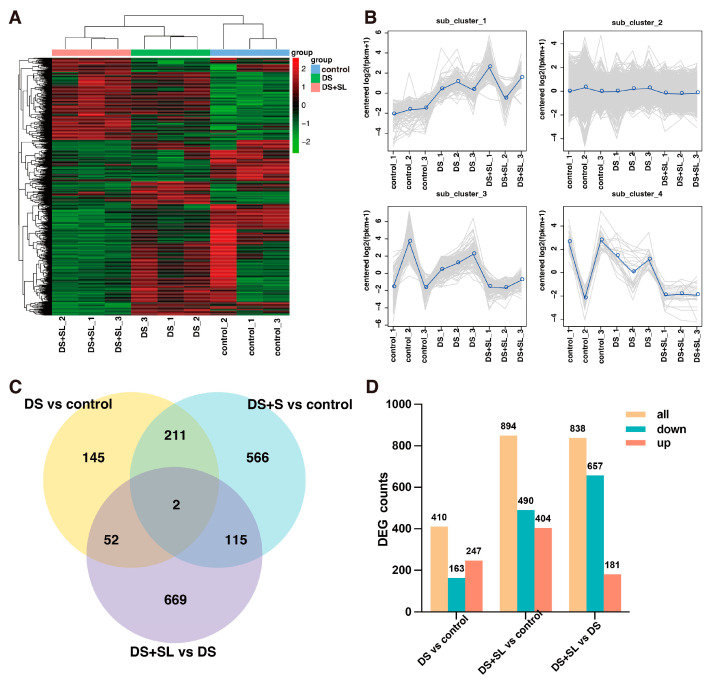
DEGs among three groups under different treatments. (**A**) Heatmap and cluster analysis expression of DEGs, (**B**) magnified regions of 4 subclusters of RNA–seq data. The gray lines represent the relative corrected gene expression levels of genes and the blue lines indicate consensus on relative corrected gene expression levels of all genes, (**C**) Venn diagram of three groups’ DEGs, and (**D**) number of DEGs.

**Figure 5 antioxidants-12-02019-f005:**
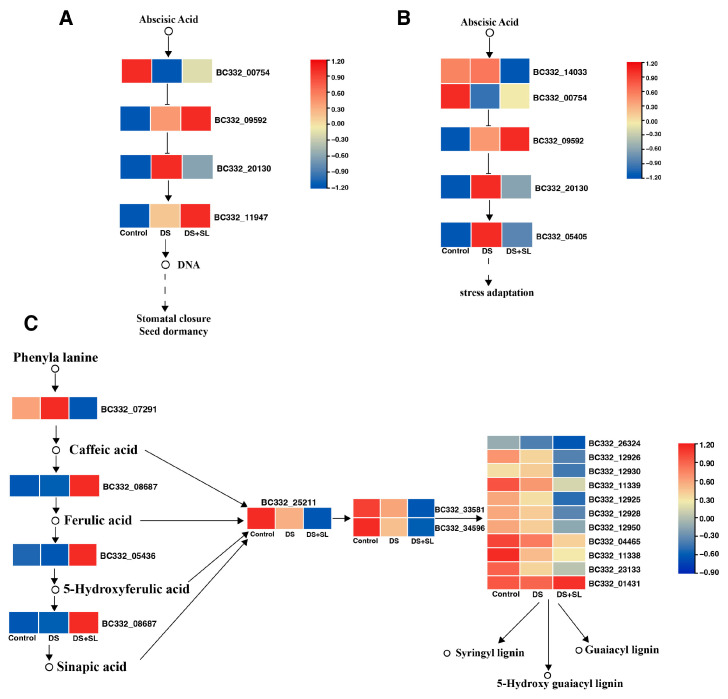
Enrichment pathways related to drought and SL treatments. (**A**) Plant hormone transduction, (**B**) MAPK signal, and (**C**) phenylpropanoid biosynthesis pathway. The heatmap colors show the log2 values of FPKM of control, DS, and DS + SL, from left to right.

**Figure 6 antioxidants-12-02019-f006:**
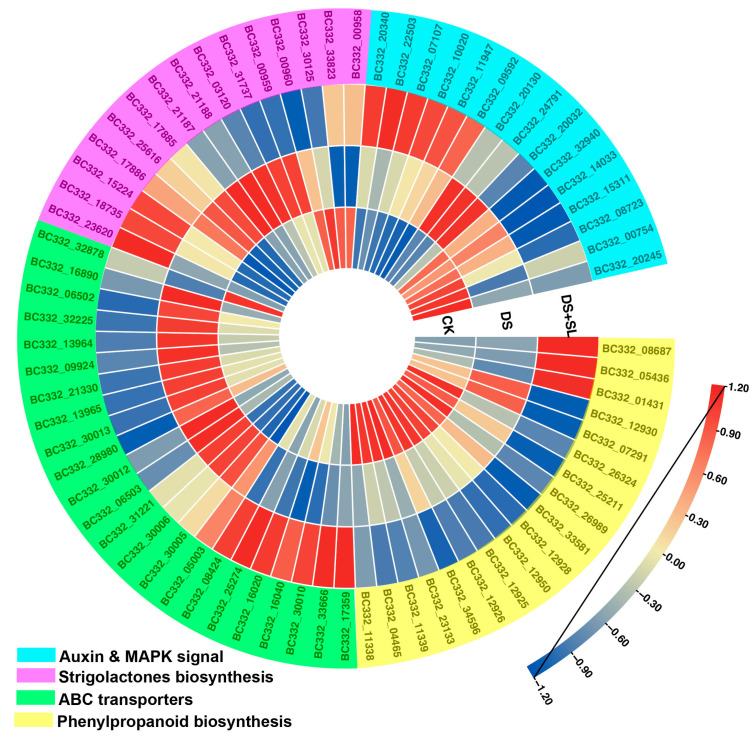
Expression of DEGs related to drought and SL treatments in control, DS, and DS + SL, respectively. Heatmap colors show the log2 values of FPKM.

**Figure 7 antioxidants-12-02019-f007:**
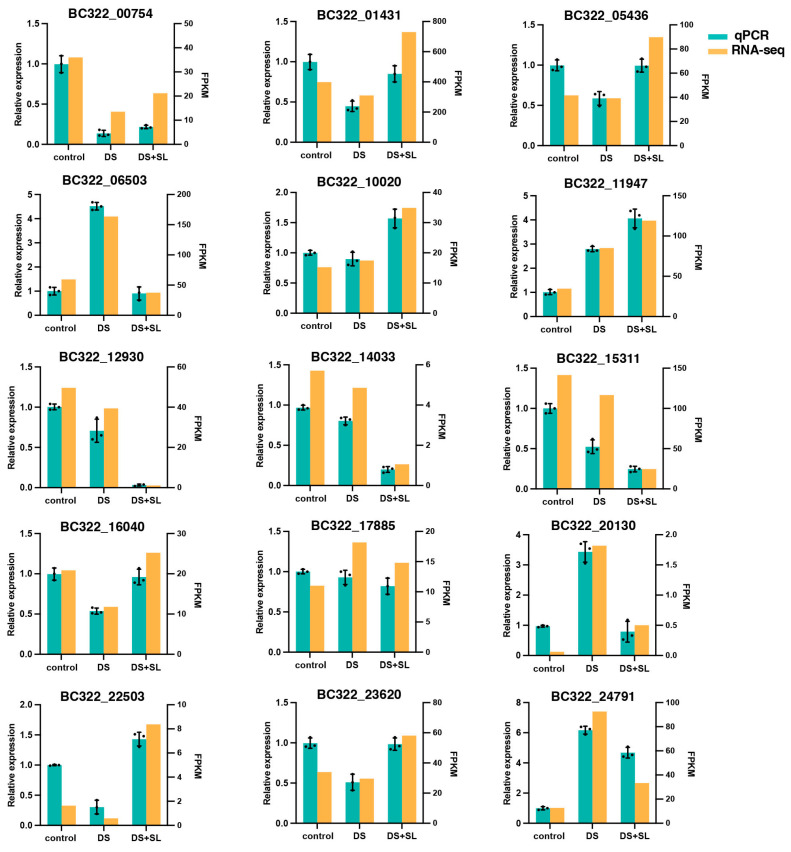
qRT-PCR validation of the relative expression levels of DEGs. Data are mean values and standard errors; the black dot indicates 3 biological replicates.

**Figure 8 antioxidants-12-02019-f008:**
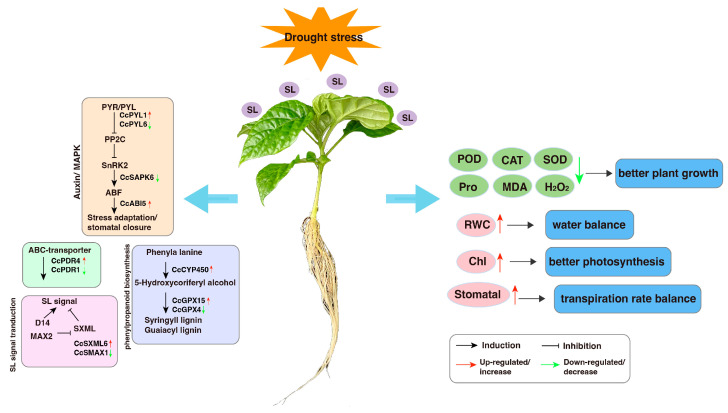
Summary of the mechanism to tolerance to drought stress induced by SL in pepper seedlings.

## Data Availability

The original data can be found in NCBI (PRJNA957909).

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
