# Peer review of "Physiological and Transcriptome Analysis of the Effects of Exogenous Strigolactones on Drought Responses of Pepper Seedlings"

_antioxidants, 2023, doi:10.3390/antiox12122019_

Round 1

Reviewer 1 Report

Comments and Suggestions for Authors

In the present manuscript Shu and collaborators describe the effects of exogenous strigolactons on pepper seedlings in drought tolerance under physiological and transcriptional perspectives.

The experiments are sound and well performed but I need some discussion about the interpretation of the results. In fact, usually SL mediate stomatal closure (which makes sense, considering their role in drought tolerance) but you indicate (section 3.2 and Fig 2A-B-C) that they alleviate stomatal closure as a result of SL activity that relieves symptoms allowing for a reopening of the stomata.

The work was remarkable and the discussion introduced interesting aspects; I personally would be interested in the perturbations in gene expression led by the supplement of strigolactons: in the last years some papers have addressed this question using similar approaches (Xu et al, 2023; Nisa et al, 2023; Krukowski et al, 2022, Daszkowska-Golec et al, 2023) so I wonder if such results can be confirmed in your settings and whether they may help you to draw a more comprehensive picture of such complex mechanism.  

Section 3.5: you should indicate the groups you sequenced. 

Line 334, ... and related Fig 4B: how did you originate the 4 clusters?

Figure 1: the picture depicting the root system in the 3 treatments is not referenced; moreover some statistics depicting (eventual) significant difference will be useful, as you did for Fig. 2.

Figure 4C-D: the sum of DEGs for DS+SLvsCK are different

Table 2: there is no table 2 shown in the manuscript, and there is no Table 1 at all.

https://doi.org/10.1016/j.scienta.2022.111800

https://doi.org/10.1093/pcp/pcac058

https://doi.org/10.1007/s10142-023-01151-8

https://doi.org/10.1186/s12870-023-04450-1

https://doi.org/10.7717/peerj.13551

https://doi.org/10.3389/fpls.2023.1186718

Comments on the Quality of English Language

English quality is overall good, and I could not identify major issues with the exception of the "Material and Methods" section that needs an help. Other than that, I would recommend a second thorough reading to improve readability (as an example, in line 100 the verbs should be used in their passive form as done in the other sections of the manuscript).

Author Response

  1. The experiments are sound and well performed but I need some discussion about the interpretation of the results. In fact, usually SL mediate stomatal closure (which makes sense, considering their role in drought tolerance) but you indicate (section 3.2 and Fig 2A-B-C) that they alleviate stomatal closure as a result of SL activity that relieves symptoms allowing for a reopening of the stomata.

Response back: Thanks for the reviewer’s keen observation. Based on microscopic structure observations, it has been found that SL alleviates stomatal closure, but its specific mechanism still needs further exploration in the future.

  1. The work was remarkable and the discussion introduced interesting aspects; I personally would be interested in the perturbations in gene expression led by the supplement of strigolactons: in the last years some papers have addressed this question using similar approaches (Xu et al, 2023; Nisa et al, 2023; Krukowski et al, 2022, Daszkowska-Golec et al, 2023) so I wonder if such results can be confirmed in your settings and whether they may help you to draw a more comprehensive picture of such complex mechanism.  

Response back: Thanks for the reviewer's keen suggestion. we will consult the references in recent years as much as possible and need to do further research.

  1. Section 3.5: you should indicate the groups you sequenced. 

Response back: The relevant information of the sequencing group is included in the supplementary document.

  1. Line 334, ... and related Fig 4B: how did you originate the 4 clusters?

Response back: We used the H-cluster method to take log2 (FPKM+1) for the expression level of differential genes and perform clustering after center correction. Divide differential genes into several clusters, and genes in the same cluster exhibit similar expression levels under different treatment conditions.

  1. Figure 1: the picture depicting the root system in the 3 treatments is not referenced; Moreover some statistics depicting (eventual) significant difference will be useful, as you did for Fig. 2.

Response back: We have provided a comprehensive summary description.

  1. Figure 4C-D: the sum of DEGs for DS+SLvsCK are different

Response back: Thanks for the reviewer’s keen observation. We have corrected the mistakes (New version Figure 4D).

  1. Table 2: there is no table 2 shown in the manuscript, and there is no Table 1 at all.

Response back: We have corrected the mistakes (New version Line 642).

  1. English quality is overall good, and I could not identify major issues with the exception of the "Material and Methods" section that needs a help. Other than that, I would recommend a second thorough reading to improve readability (as an example, in line 100 the verbs should be used in their passive form as done in the other sections of the manuscript).

Response back: Thanks for the reviewer’s keen observation. We have revised (New version Line 366).

Reviewer 2 Report

Comments and Suggestions for Authors

In this paper, the authors conducted a physiological and transcriptome analysis, to uncover

the mechanisms through which Strigolactones (SL) mitigate the effects of drought stress on pepper seedlings.

There is a nice description of the physiological responses of plants pre treated with SL in response to drought. The SL pretreatment does alleviate the stress.

POD and SOD is still high in SL+SD (similar too SD) showing the plants are still dealing with stress.

I do not approve the use of “CK” for control. CK is a common abbreviation for cytokinin. I see no “k” in control. so use Ct or, better, write “control”!

why are they no quantitation of stomata aperture?

why is there no SL treatment alone?

The title is not good.

I would suggest

Physiological and Transcriptome Analysis of the effects of Exogenous Strigolactones on drought responses of Pepper Seedlings

line 18 abbreviation SL not explained

line 17. these categories are enriched in DS and DS+SL. So not characteristics of SL response?

line 29? treated with SL and DS?

lie 30. Not clear. down regulated I what conditions compared to what?

line 31. To be honest I see no mechanism. maybe you should express it.

line 47: Is “thus captivating globally” correct English?

line 60: what do you mean by “exogenous SL has been initiated”??

lines 78/79: very vague. Here it should be the take home message.

line 85: wht were “renewed”? the seedlings?? or the medium?

line 88: what is uM?

why use “CK” for control?? I see no k in control. so use Ct or better “control”

line 92: what is “it” for??

line 112: what is “it” for??line 113: M&M should be written in passive form

line 114: blocks are left etc…

line 115: “were dried”

line 116: “observe”: is it an order to the reader???

line 120/121: rewrite correctly

line 137: what do you consider to be normal?

line 151 qPCRs were analyzed

line 152 maybe delete “Using the previous protocol method,”

Figure 1: write “Control” and no “CK”!!!

line 184: “In the SL pretreatment and DS group,” why do not you write “in plant pretreated with SL and then submitted to DS, stomatal closure was les important than in plants treated with DS”??

why are they no quantitation of stomata aperture

Figure 2: write “Control” and no “CK”!!!

Figure 3: write “Control” and no “CK”!!!

Figure 4: write “Control” and no “CK”!!!

Figure 5: write “Control” and no “CK”!!!

Figure 6: write “Control” and no “CK”!!!

how are we supposed to read figure 5? what are the genes encoding? in the 3 column color rows, what is frst, two and third column?

Figure 4. Panel D. what is “comprare”?

what is the point of the circle representation? a basic heatmap would be clearer.

Comments on the Quality of English Language

few editng necesary

Author Response

  1. POD and SOD is still high in SL+SD (similar too SD) showing the plants are still dealing with stress.

Response back: Thanks for the reviewer’s keen observation. Under drought stress, SL treatment significantly reduced the activity of POD, but the difference in SOD was not significant, and further exploration is needed.

  1. I do not approve the use of “CK” for control. CK is a common abbreviation for cytokinin. I see no “k” in control. so use Ct or, better, write “control”!

Response back: Thanks for the reviewer’s suggestion. We have revised.

  1. why are they no quantitation of stomata aperture?

Response back: The analysis results are presented in supplementary materials.

  1. why is there no SL treatment alone?

 Response back: The experimental design only considered the impact of SL on drought stress, without considering the impact of SL on plant growth. We will consider adding this treatment group in subsequent experiments.

  1. The title is not good. I would suggest “Physiological and Transcriptome Analysis of the effects of Exogenous Strigolactones on drought responses of Pepper Seedlings”

Response back: As per the reviewer’s suggestion, we have revised the title.

  1. line 18 abbreviation SL not explained

Response back: As per the reviewer’s suggestion, we have revised. (New version Line 17)

  1. line 17. these categories are enriched in DS and DS+SL. So not characteristics of SL response?

line 29? treated with SL and DS?

Response back: Thanks for the reviewer’s observation. In the different treatments, the pathway was significantly enriched, but in different treatments, some genes on the enriched pathway were upregulated and some were downregulated, so we speculate that it may be related to SL.

  1. line 30. Not clear. down regulated I what conditions compared to what?

Response back: Thanks for the reviewer’s observation. We have rewritten. (New version Line 30).

  1. line 31. To be honest I see no mechanism. maybe you should express it.

Response back: As per the reviewer’s suggestion, we have revised.

  1. line 47: Is “thus captivating globally” correct English?

Response back: As per the reviewer’s suggestion, we have revised. (New version Line 40).

  1. line 60: what do you mean by “exogenous SL has been initiated”??

Response back: As the reviewer’s observation, Previous studies have shown that exogenous SL has begun to be applied in drought stress and achieved certain results.

  1. lines 78/79: very vague. Here it should be the take home message.

Response back: As per the reviewer’s suggestion, we have revised.

  1. line 85: wht were “renewed”? the seedlings?? or the medium?

Response back: As per the reviewer’s suggestion, we have revised.

  1. line 88: what is uM?

Response back: Thanks for the reviewer’s observations. We have revised the mistake. (New version Line 107).

  1. why use “CK” for control?? I see no k in control. so use Ct or better “control”

Response back: Thanks for the reviewer suggestion. We have revised.

  1. line 92: what is “it” for??

Response back: We have revised. (New version Line 379)

  1. line 112: what is “it” for??line 113: M&M should be written in passive form

Response back: We have revised. (New version Line 387)

  1. line 114: blocks are left etc…

Response back: We have revised. (New version Line 378)

  1. line 115: “were dried”

Response back: We have revised. (New version Line 382)

  1. line 116: “observe”: is it an order to the reader???

Response back: We have revised. (New version Line 383)

  1. line 120/121: rewrite correctly

Response back: Thank you for the suggestion, we have revised.

  1. line 137: what do you consider to be normal?

Response back: We select seedlings that grow normally and are free from pests and diseases.

  1. line 151 qPCRs were analyzed

Response back: Thank you for the suggestion, we have revised. (New version Line 433).

  1. line 152 maybe delete “Using the previous protocol method,”

Response back: Thank you for the suggestion, we have revised. (New version Line 434).

  1. Figure 1: write “Control” and no “CK”!!!

Response back: Thank you for the suggestion, we have thoroughly revised.

  1. line 184: “In the SL pretreatment and DS group,” why do not you write “in plant pretreated with SL and then submitted to DS, stomatal closure was les important than in plants treated with DS”??

Response back: Thank you for the suggestion, we have thoroughly revised (New version Line 450).

  1. why are they no quantitation of stomata aperture

 Response back: The analysis results are presented in supplementary materials (Figure S1).

  1. Figure 2: write “Control” and no “CK”!!!

Figure 3: write “Control” and no “CK”!!!

Figure 4: write “Control” and no “CK”!!!

Figure 5: write “Control” and no “CK”!!!

Figure 6: write “Control” and no “CK”!!!

Response back: Thank you for the suggestion, changes have been made to the new version.

  1. how are we supposed to read figure 5? what are the genes encoding? in the 3 column color rows, what is frst, two and third column?

Response back: Thank you for the observations. The heatmap colors showed the FPKM expression values of Control, DS, and DS+SL from left to right, respectively. (New version Line 605)

  1. Figure 4. Panel D. what is “comprare”?

 Response back: Thank you for your observation. Compare represents a comparison between three different treatment groups, which may be inappropriate and has been modified in the new version.

  1. what is the point of the circle representation? a basic heatmap would be clearer.

Response back: Due to the large number of genes, a basic a regular rectangle heatmap will occupy too much space, so circular ones are used.

Round 2

Reviewer 1 Report

Comments and Suggestions for Authors

I thank the authors for having taken into consideration some of the points I adressed; however there are some more that, in my view, were not sorted out so I ask the authors to reply. 

lines 92-93: please add a more detailed protocol for SL application (sprayed onto leaves? how often?)

line 142: what do you mean with "mixed sampling"? it seems that you pooled tissues fron various conditions together.

section 2.6: as you did not explain elsewhere how you generated Fig. 4B, please add it here.

Lines 252-253: I think the phrase is partial (no verb and no explanations).

Line 255: "were significantly enriched": please explain the statistics used to support this claim.

Discussion: can you comment the results you got regarding SL-mediated opening of the stomata? I ask this since other literature reports differently (Lv et al, 2018, Zhang et al, 2018, Kalliola et al 2020, etc).

Moreover, SL acts in concert with ABA in a series of responses including stomata: did you identify ABA-related genes among DEGs?

Other than that, since other papers followed an approach similar to yours (references added in previous review), you should discuss your results with their respect and see whether differences can be found. This would also help you to model your Fig. 8. 

Figure 1: the picture of the 3 rooting systems is not included in the caption: please do it. Moreover, you should describe what the dots and the letters represent in the panels. If statistics was used, it should be added in order to justify your claim in line 169 ("DS significantly inhibited...").

Comments on the Quality of English Language

Please again revise the manuscript as some editing is still necessary (ex.: line 42: ('environmental factors').

Author Response

Reviewer 1

I thank the authors for having taken into consideration some of the points I adressed; however, there are some more that, in my view, were not sorted out so I ask the authors to reply. 

lines 92-93: please add a more detailed protocol for SL application (sprayed onto leaves? how often?)

Response back: Thanks for the reviewer’s keen observation. Yeswe spayed onto the leaves, twice a day, and continuous three days.

line 142: what do you mean with "mixed sampling"? it seems that you pooled tissues from various conditions together.

Response back: Sorry for our unclear expression. The mixed sample indicates that 3-5 plant leaves are taken from each treatment as a sample. Not a mixture of different conditions. We have revised (New version Line 142)

section 2.6: as you did not explain elsewhere how you generated Fig. 4B, please add it here.

Response back: Each subset of the clustering line graph, where the value of differential genes is the union of all comparative combinations of differential genes, and the expression level of differential genes in the FPKM expression matrix of each sample is taken as log2 (fpkm+1) and centralized for correction. The horizontal axis represents the sample name, and the vertical axis represents the expression value after logarithmic center correction. The gray lines in each subgraph represent the relative corrected gene expression levels of genes in a cluster under different experimental conditions, while the blue lines represent the average relative corrected gene expression levels of all genes in the cluster under different experimental conditions.

Lines 252-253: I think the phrase is partial (no verb and no explanations).

Response back: Thanks for the reviewer’s keen observation. We have revised (New version Line 253).

Line 255: "were significantly enriched": please explain the statistics used to support this claim.

Response back: Firstly, KEGG (Kyoto Encyclopedia of Genes and Genomes) is a comprehensive database that integrates genomic, chemical, and systemic functional information. The threshold for significant enrichment of KEGG pathway is padj < 0.05. In this study, we used clusterProfiler software to conduct KEGG pathway enrichment analysis on the differential gene set. We conducted enrichment analysis on all differential genes in each differential comparison combination to obtain a set of enriched pathways. Furthermore, based on previous studies, we identified pathways that may be related to stress for further analysis, with a focus on significantly differentially expressed genes in these pathways. Our statement may be inappropriate and we have been revised. Thank you for your question.

Discussion: can you comment the results you got regarding SL-mediated opening of the stomata? I ask this since other literature reports differently (Lv et al, 2018, Zhang et al, 2018, Kalliola et al 2020, etc).

Response back: Our hypothesize that SL may play a crucial role in regulating plant water balance, leading to the reopening of stomata. Previous studies have shown that many genes involved in stomatal opening are downregulated by ABA and water stress, but it may be necessary for plants to maintain the expression of some genes involved in stomatal opening in protective cells to maintain stomatal opening at specific levels and subsequently balance CO2 inflow and water loss under water scarcity conditions (Ding et al, 2014). This is a complex molecular mechanism that still needs further research.

Moreover, SL acts in concert with ABA in a series of responses including stomata: did you identify ABA-related genes among DEGs?

Response back: We have also found a close relationship between drought stress and ABA and ROS signal in relevant literature. Unfortunately, no relevant genes were found in this study, but they can be our focus in the future. Thank you for the question.

other papers followed an approach similar to yours (references added in previous review), you should discuss your results with their respect and see whether differences can be found. This would also help you to model your Fig. 8. 

Response back: Thank you for good suggestion. We have already discussed our results with previous published reports. Generally, Figure 8 described the mechanism of whole study results. We have tried to present my all results summery in figure. We will consider your opinion in our upcoming manuscript, thank you for valuable suggestion.

Figure 1: the picture of the 3 rooting systems is not included in the caption: please do it. Moreover, you should describe what the dots and the letters represent in the panels. If statistics was used, it should be added in order to justify your claim in line 169 ("DS significantly inhibited...").

Response back: Thanks for the reviewer’s keen suggestion. We have revised the Figure 1(New version Line 179) and describe what the dots and the letters represent (New version Line 183).

Reviewer 2 Report

Comments and Suggestions for Authors

The authors took into consideration some of my remarks.

I still consider it is a pity there is no data for SL treatment alone

In the future, please always prepare this sample.

Be more precise when you deal with absolute values (as opposed to ratio between two conditions)

I consider that some titles are still unprecise, and that some captions would need more details.

Line 124. Grounded

Line 138, soluble what ?

I think you should put the S1 data in figure 2

Line 227: add a space between in and control

Figure 5. as I already asked, please add for each column what is it we see . You write in caption “The heatmap colors showed the FPKM expression values of control, DS, and DS+SL

from left to right, respectively.” Are they log2 values, or absolute values?… I guess for instance it is DS+SL vs DS, but how can we be sure since you do not write it.

Line 268: “The heatmap colors showed the FPKM expression values… “ what is FPKM? Is it not expression of genes? Are they absolute values ?

Same remark for figure 6. When you write DS+SL , you mean DS+SL vs DS?

What is “enrichment pathways”?

Supplemental.

S1 what is “percentage of stomatal”?

S2 caption: add space before after after “vs”

Idem for S3

S3 title is “enrichment” but it seems you deal on number of genes. It should be compared to the presence of each category in the whole genome, to see really “enrichment”

Comments on the Quality of English Language

I think there is still some issues with the language.

Author Response

The authors took into consideration some of my remarks.

I still consider it is a pity there is no data for SL treatment alone

In the future, please always prepare this sample.

Be more precise when you deal with absolute values (as opposed to ratio between two conditions)

I consider that some titles are still unprecise, and that some captions would need more details.

 Response back: Thank you for your concern, in this assignment, we have only selected 3 treatment. we will consider your suggestion for our upcoming work.

Line 124. Grounded

Response back: Thanks for the reviewer’s keen observation. We have revised (New version Line 124).

Line 138, soluble what?

Response back: Thanks for the reviewer’s keen suggestion. We have revised in the new version. (New version Line 138)

I think you should put the S1 data in figure 2

Response back: Thanks for the reviewer’s keen suggestion. We have revised in the new version (Figure 2D).

Line 227: add a space between in and control

Response back: Thanks for the reviewer’s keen suggestion. We have revised in the new version.

Figure 5. as I already asked, please add for each column what is it we see. You write in caption “The heatmap colors showed the FPKM expression values of control, DS, and DS+SL from left to right, respectively.” Are they log2 values, or absolute values?… I guess for instance it is DS+SL vs DS, but how can we be sure since you do not write it.

 Response back: Thank you for correction. We have drawn the heat map with log2 values of FPKM. We have rewritten the Figure 5.

Line 268: “The heatmap colors showed the FPKM expression values… “what is FPKM? Is it not expression of genes? Are they absolute values?

Response back: FPKM (expected number of Fragments Per Kilobase of transcript sequence per Millions base pairs sequenced). It refer to the number of paired reads per million fragments per thousand base lengths of a certain gene, which takes into account the effects of sequencing depth and gene length on fragment count. It is currently a commonly used method for estimating gene expression levels. We have explained in the legends.

Same remark for figure 6. When you write DS+SL, you mean DS+SL vs DS?

 Response back: This figure not belong to compression treatments such as DS+SL vs DS or Control vs DS. This heat map described the difference of each treatment. such control, DS belong drought stress treatment, and DS+SL belong combined application of drought stress and SL use as pretreatment. Generally, we want to described the difference among treatments.

What is “enrichment pathways”?

 Response back: The enrichment pathway is based on our KEGG enrichment analysis of differential genes. In addition, we combined the results of previous studies to screen out pathways that may be related to drought stress. The enrichment pathway is based on our KEGG enrichment analysis of differential genes. In addition, we combined the results of previous studies to screen out pathways that may be related to drought stress. We presented the results in section 3.6 and Figure 5. Thank you for your questions.

Supplemental.

S1 what is “percentage of stomatal”?

S2 caption: add space before after “vs”

Idem for S3

Response back: Percentage of stomatal means the percentage of open stomatal area to the total stomatal area in different treatments, in order to further illustrate the degree of stomatal opening in different treatments. We want to express the percentage of open stomatal area to total stomatal area. But this statement can be confusing, so we have revised. (New version Figure 2D). We have already added the space according to your highlighted point. Thank you for the valuable suggestions.

S3 title is “enrichment” but it seems you deal on number of genes. It should be compared to the presence of each category in the whole genome, to see really “enrichment”

Response back: We used clusterProfiler software to perform KEGG functional enrichment analysis on differential gene sets Perform enrichment analysis on all differentially expressed genes in each differential comparison combination. Under each enrichment directory, ALL; UP; DOWN represents all differentially expressed genes for each differential comparison combination; Upregulation of differentially expressed genes; Enrichment results of downregulated differentially expressed genes.

Round 3

Reviewer 1 Report

Comments and Suggestions for Authors

I thank the authors for having taken into consideration my notes and for the discussion that followed.

I think that now the manuscript can be published; I only ask some small things.

The first concerns -again- lines 92-93: I appreciated your explanation, but you have to write in onto the manuscript, in order to let the readers realize that leaves were sprayed and not, for instance, dipped into a SL solution.

section 2.6: I thanks the authors for the explanation, but probably I was unclear: I asked you to add it either on section 2.6 or into the caption of Fig. 4B in order to let the readers understand the protocol you used.

Last thing, regarding your results in stomata regulation: I still believe that, since you reported a behavior that is different from published literature, you should discuss it: what you wrote for answering my question in the last revision could be enough: your ref 32 does not describe the effect of SLs on stomata: if your previous reports described this phenomenon, you can report it in the discussion as well.

Author Response

The first concerns -again- lines 92-93: I appreciated your explanation, but you have to write in onto the manuscript, in order to let the readers realize that leaves were sprayed and not, for instance, dipped into a SL solution.

Response back: Thanks for the reviewer’s keen suggestion. We have revised (New version Line 92-93).

section 2.6: I thanks the authors for the explanation, but probably I was unclear: I asked you to add it either on section 2.6 or into the caption of Fig. 4B in order to let the readers understand the protocol you used.

Response back: We have revised (New version Line 150-153 and Line 253-255).

Last thing, regarding your results in stomata regulation: I still believe that, since you reported a behavior that is different from published literature, you should discuss it: what you wrote for answering my question in the last revision could be enough: your ref 32 does not describe the effect of SLs on stomata: if your previous reports described this phenomenon, you can report it in the discussion as well.

Response back: We have revised (New version Line 358-360).